# Application of naïve Bayesian approach in detecting reproducible fatal collision locations on freeway

**Eui-Jin Kim**[1], **Oh Hoon Kwon**[2]*, **Shin Hyoung Park**[3], **Dong-Kyu Kim**[4], **Koohong Chung**[5]

**1** Department of Civil and Environmental Engineering, Seoul National University, Seoul, Republic of Korea,
**2** Department of Transportation Engineering, College of Engineering, Keimyung University, Daegu, Republic of Korea, **3** Department of Transportation Engineering, University of Seoul, Seoul, Republic of Korea,
**4** Department of Civil and Environmental Engineering and Institute of Construction and Environmental Engineering, Seoul National University, Seoul, Republic of Korea, **5** School of Civil, Environmental and Architectural Engineering, Korea University, Seoul, Republic of Korea

☯ These authors contributed equally to this work.
* ohoonkwon@kmu.ac.kr

**Data Availability Statement:** All relevant data based on SWITRS database are within the paper and its Supporting Information files.

**Funding:** Dong-Kyu Kim was supported by the Basic Science Research Program through the National Research Foundation of Korea (NRF)

## Abstract

Detecting high-collision-concentration locations based solely on collision frequency may produce different results compared to those considering the severities of the collisions. In particular, it can lead government agencies focusing sites with a high collision frequency while neglecting those with a lower collision frequency but a higher percentage of injury and fatal collisions. This study developed systematic ways of detecting reproducible fatal collision locations ($R$) using the naïve Bayes approach and a continuous risk profile (CRP) that estimates the true collision risk by filtering out random noise in the data. The posterior probability of fatal collisions being reproducible at a location is estimated by the relationship between the spatial distribution of fatal-collision locations (i.e., likelihood) and the CRP (i.e., prior probability). The proposed method can be used to detect sites with the highest proxy measure of the posterior probability (PMP) of observing $R$. An empirical evaluation using 5-year traffic collision data from six routes in California shows that detecting $R$ based on the PMP outperform those based on the SPF-based approaches or random selection, regardless of various conditions and parameters of the proposed method. This method only requires traffic collision and annual traffic volume data to estimate PMP that prioritize sites being $R$ and the PMPs can be compared across multiple routes. Therefore, it helps government agencies prioritizing sites of multiple routes where the number of fatal collisions can be reduced, thus help them to save lives with limited resources of data collection.

## Introduction

The Fixing America's Surface Transportation Act [1] requires that the Highway Safety Improvement Program (HSIP) [2] and federal aid programs establish data-driven and

funded by the Ministry of Science, ICT & Future Planning (2019R1H1A1080045) and Oh Hoon Kwon was supported by the National Research Foundation of Korea (NRF) grant funded by the Korea government (MSIT) (No. NRF-2017R1C1B5017592).(https://www.nrf.re.kr/eng/index) The funders had no role in study design, data collection and analysis, decision to publish, or preparation of the manuscript.

**Competing interests:** The authors have declared that no competing interests exist.

performance-based approaches to reduce traffic fatalities and serious injuries on all public roadways. Under HSIP, state agencies that receive federal funding are required to develop a statewide coordinated plan to improve the safety of their roadways through the Strategic Highway Safety Plan (SHSP) [3]. In developing guidelines for the SHSP, state agencies are mandated under Title 23, U.S.C. Section148 HSIP (d)(1)(B) (Public Law 109–59) to consider the locations of fatalities and serious injuries. Title 23, U.S.C. Section 148 HSIP also states that the state agency must utilize a safety-data collection system with a method for detecting sites with a high frequency of fatal and serious injuries.

HSIP provides no explicit guidelines for determining high collision concentration locations (HCCL) considering severity. However, by requiring state agencies to consider the locations of fatal collisions, it is implicitly assumed that these locations are reproducible. When this assumption does not hold, it can result in the suboptimal allocation of limited government resources as the method used by a government agency to detect HCCLs essentially dictates how the agency's resources are allocated to improve the safety of the roadway system it manages.

Fig 1 shows the annual collision trend in California freeway from 2005 to 2017. Since 2005, the number of collisions has declined until 2013 and then increases again to 2017. In comparison between 2005 and 2017, the number of total collisions was quite higher in 2017, the ratio of injury collisions was slightly higher in 2017, and the ratio of the fatal collision was slightly lower in 2017 [4]. This trend indicates that both fatal and injury collisions are still issues to be addressed in spite of advances in vehicle safety systems, roadway design, and various traffic safety-countermeasures [5]. The investigation of every site where fatal collisions occur can be cost-prohibitive, considering that state agencies often struggle to cope with the long list of sites to be investigated for safety improvement [6]. As such, it is important to differentiate the sites where fatal collisions are likely to occur in the near future from those where fatal collisions are not likely to occur again.

If the percentages of fatal, serious injury, and property damage only (PDO) collisions remain unchanged or comparable across sites, the detection of HCCLs based solely on collision frequency may produce results comparable to those for HCCLs after weighting collisions with respect to severity. However, in practice, this is often not the case as sites that are plagued by recurrent freeway bottlenecks can report a high frequency of PDO collision but a low frequency of fatal and serious injuries. As traffic moves at slow speeds, collisions occurring in congested traffic conditions typically involve PDO or minor injuries. In contrast, collisions that occur while traffic is moving in a free flow state can result in a higher percentage of serious and even fatal injuries. Therefore, a procedure detecting HCCL that can differentiate the severity levels in traffic collision data must be used to efficiently reduce the number of fatal collisions.

The Federal Highway Administration [7] and California Department of Transportation (Caltrans) [8] provides general guidelines in detecting the HCCLs in the Highway Safety Manual (HSM), based on a safety performance function (SPF) that statistically predicts the expected frequency of collisions. HCCLs can be determined by comparing traffic collision data obtained for pre-defined short roadway segments, i.e., 0.1–1.5 km for homogeneous sections regarding the average annual daily traffic (AADT) [9, 10] with the expected collision frequency estimated using the SPF. While the SPFs estimate the average collision frequency for all collisions, the HSM suggests a procedure to estimate the collision frequency by crash severity. For instance, the number of fatal or injury collisions can be estimated by separating the estimated total crash frequencies as distributions of crash severity calibrated by local data. SPFs directly calibrated by fatal and injury (FI) collisions [11, 12] or fatal collisions [13] have also been developed in previous studies. Although such procedures can be used to estimate the expected

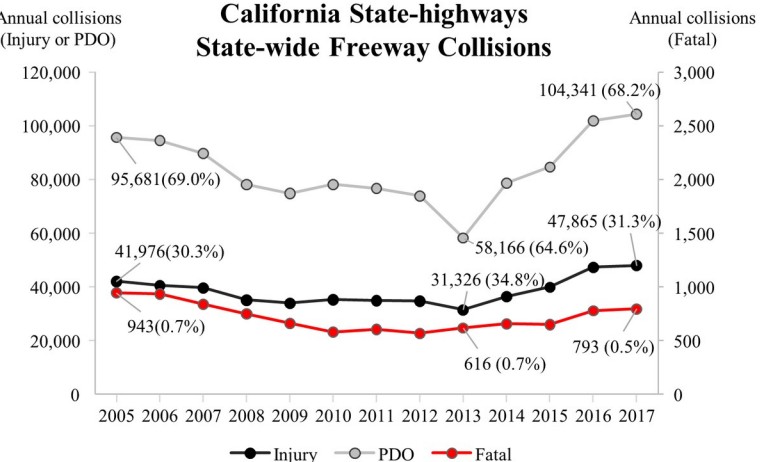

**Fig 1. Annual collision trend in California from 2005 to 2017.**

number of fatal collisions in sites that are segmented by roadway attributes, the infrequency of fatal collisions causes a bias in safety measures, such as the potential for safety improvement (PSI), due to the extremely low number of expected and observed values. Therefore, compared with the PSI values for the FI or total number of collisions, which are well established and for which procedures have been verified [7, 14], the reliability of the PSI value for fatal collisions may be significantly worse.

To remedy this situation, researchers [15–17] have proposed a method for assigning different weights to collisions based on their severity level to identify sites that may experience low collision frequency but a high percentage of serious injuries. These authors have proposed weighting collision frequencies with respect to collision severity based on the associated economic loss. Blincoe et al. reported that the average economic loss associated with fatal collisions in 2000 was 1,330 times higher than that of PDO collisions [18]. Therefore, weighting collisions based on their associated economic loss can result in problems when some of the fatal collisions that occur randomly over extended roadway segments are solely the fault of the driver (i.e., driving under the influence (DUI) and suicide). When this occurs, it can increase both the false-positive rate (i.e., flagging a site as an HCCL when it is not) and the false-negative rate (i.e., not detecting a true HCCL). Robustness and consistency are important performance measures in the identification of HCCLs [19], and a comparative analysis of HCCL identifications revealed that safety evaluations for which the severity of collisions is weighted based on the associated economic loss are subject to inconsistency over repeated observation periods [16].

To address these issues, in the present study, systematic methods for detecting reproducible fatal collision locations were developed using Bayes' rule and the continuous risk profile (CRP) [20–22]. While the Bayes' rule have been used for SPF-based approaches to improve the model performance using past evidence from other SPF as a prior [23], this study used the CRP as a prior for detecting reproducible fatal collision locations. The performance of the proposed method was empirically evaluated using traffic collision data from freeway routes in California. This method can assist government agencies in improving the safety of traffic corridors even when the end-points of routes to be investigated have been determined by other external factors (such as funding requirements, local jurisdiction boundaries, or other conflicting projects). The proposed method also can help agencies to prioritize funding for reducing the number of fatal collisions by identifying reproducible fatal collision locations. The findings from our empirical evaluation show strong promise for the systematic detection

of reproducible fatal collision locations. The following section describes the data used in this study. A description of the proposed method together with the results obtained by its application to the empirical data are presented in the next section. Subsequently, we report the findings based on the evaluation of the proposed method. Brief concluding remarks and future research plans are provided at the end of the paper.

## Site and data

All the traffic collision data used in this study are based on State Wide Integrated Traffic Records Systems (SWITRS) that are open-source database published by California Highway Patrol [24]. The SWITRS database records all vehicle collisions occurred on a public road-way. Among the traffic collision in SWITRS database, Traffic Accident Surveillance and Analysis System (TASAS) collected the collision data that occurred on a road that is managed by Caltrans, and this study used the TASAS collision data. The traffic collision data used was collected between 2004 and 2008 from 390 miles of six freeway routes in the San Francisco bay area, including I-80W, I-80E, I-580W, I-580E, I-880N, and I-880S. These routes share similar features, i.e., they are all multi-lane, interstate, urban freeways with high traffic volume. The purpose of the study is to evaluate the reproducibility of fatal collision locations on freeways based on the collision occurrence data and not depending on the influential factors that are changed over time such as vehicle safety systems, roadway conditions, and traffic conditions. Therefore, to evaluate our method by comparing carefully calibrated previous method, we used these datasets applied in the previous studies [14] rather than recent data. Over the five years of the study, 49,159 collisions occurred on the six routes and those data were uploaded as the supporting information in **S1 Dataset**.

Table 1 shows the total number of collisions for each route by collision severity and year. The length of each route ranged from 50 to 80 miles. The average number of collisions per mile per year for each route varied from 15 to 32. The annual number of collisions for each route decreased over the study period. Table 1 shows the proportions of collision severity levels for each route by year. Since the percentages of fatal, injury, and PDO collisions changed over the years or across routes, HCCL identification based on the total number or number of injury collisions would be different from the identification of reproducible fatal collision locations.

Table 2 shows the results of a chi-square test conducted to determine whether there is a significant association between the proportions of collision severity levels and the year or the route. The left side of Table 2 indicates that there is a significant association between the year and proportions of collision severity levels on all routes. The right side of Table 2 also shows a statistically significant association between the routes and proportions of collision severity levels in all years. In particular, these results indicate that the proportion of fatal collisions would be varied significantly in terms of ratio from year to year in all routes, and from route to route in all years. This tendency indicates that collision severity levels should be considered in identifying the reproducible fatal collisions to evaluate the safety of the site properly.

## Methodology

The objective of the proposed method is to identify sites where fatal collisions are likely to occur within the next few years among sites where a fatal collision had already occurred. Using the naïve Bayes approach, the proposed method only evaluates the proxy measure of posterior probability (PMP) of additional fatal collisions occurring within subsequent years in the vicinities of the fatal collisions reported in the reference year (i.e., reproducible fatal collision locations). This section describes the reproducible and non-reproducible fatal collision sites and the procedure used for estimating PMP.

**Table 1. Total number of collisions for each route by collision severity and year.**

| | Route | I-80W | I-80E | I-580W | I-580E | I-880N | I-880S |
|---|---|---|---|---|---|---|---|
| | Length (mile) | 80 | 80 | 65 | 65 | 50 | 50 |
| | Average Collisions/mile/year | 29 | 25 | 22 | 15 | 32 | 30 |
| Total Collisions | 2004 | 2,715 | 2,210 | 1,494 | 1,020 | 1,747 | 1,594 |
| | 2005 | 2,461 | 2,120 | 1,538 | 1,089 | 1,679 | 1,645 |
| | 2006 | 2,421 | 1,955 | 1,481 | 1,005 | 1,556 | 1,496 |
| | 2007 | 2,245 | 1,857 | 1,485 | 972 | 1,508 | 1,469 |
| | 2008 | 1,907 | 1,763 | 1,131 | 804 | 1,401 | 1,391 |
| Fatal Collisions | 2004 | 0.2% | 0.6% | 0.4% | 0.1% | 0.3% | 0.4% |
| | 2005 | 0.6% | 0.4% | 0.3% | 1.1% | 0.4% | 0.3% |
| | 2006 | 0.4% | 0.7% | 0.4% | 0.4% | 0.3% | 0.6% |
| | 2007 | 0.6% | 0.8% | 0.1% | 0.6% | 0.2% | 0.2% |
| | 2008 | 0.4% | 0.9% | 0.7% | 0.6% | 0.8% | 0.7% |
| Injury Collisions | 2004 | 23.0% | 22.6% | 33.6% | 34.1% | 27.9% | 33.1% |
| | 2005 | 23.3% | 25.6% | 31.9% | 30.9% | 31.2% | 28.4% |
| | 2006 | 22.6% | 27.1% | 28.2% | 29.5% | 29.8% | 30.4% |
| | 2007 | 26.8% | 27.0% | 31.0% | 31.1% | 30.4% | 30.8% |
| | 2008 | 28.3% | 31.4% | 31.3% | 29.9% | 26.4% | 28.4% |
| PDO Collisions | 2004 | 76.8% | 76.8% | 66.0% | 65.8% | 71.8% | 66.4% |
| | 2005 | 76.1% | 74.0% | 67.8% | 68.0% | 68.4% | 71.2% |
| | 2006 | 77.0% | 72.3% | 71.4% | 70.1% | 69.9% | 69.0% |
| | 2007 | 72.6% | 72.2% | 68.9% | 68.3% | 69.4% | 69.0% |
| | 2008 | 71.3% | 67.7% | 68.0% | 69.5% | 72.8% | 70.9% |

## Reproducible (R) and non-reproducible (N) fatal collision site

To evaluate the performance of the proposed method, the reproducible (R) and non-reproducible (N) fatal collision sites were empirically classified based on the occurrence of additional fatal collisions in subsequent years (i.e., validation year) near where a fatal collision had occurred in the reference year. Fig 2 shows the R and N at I-80W for which the reference year is

**Table 2. Statistical test results for the proportions of collision severity levels over years and routes.**

| Chi-square test for independence | | | |
|---|---|---|---|
| $H_0$: There is no significant association between the proportion of collision severity and year 2004–2008 | | $H_0$: There is no significant association between the proportion of collision severity and routes (I-80W, I-80E, I-580W,I-580E,I-880N,I-880S) | |
| $H_1$: There is significant association between the proportion of collision severity and year 2004–2008 | | $H_1$: There is significant association between the proportion of collision severity and routes (I-80W, I-80E, I-580W,I-580E,I-880N,I-880S) | |
| I-80W | Reject $H_0$ (p-value < 0.001) | 2004 | Reject $H_0$ (p-value < 0.001) |
| I-80E | Reject $H_0$ (p-value < 0.001) | 2005 | Reject $H_0$ (p-value < 0.001) |
| I-580W | Reject $H_0$ (p-value = 0.015) | 2006 | Reject $H_0$ (p-value < 0.001) |
| I-580E | Reject $H_0$ (p-value = 0.044) | 2007 | Reject $H_0$ (p-value < 0.001) |
| I-880N | Reject $H_0$ (p-value = 0.030) | 2008 | Reject $H_0$ (p-value = 0.048) |
| I-880S | Reject $H_0$ (p-value = 0.019) | | |

*Note*: $H_0$ is rejected at the 5% significance level.

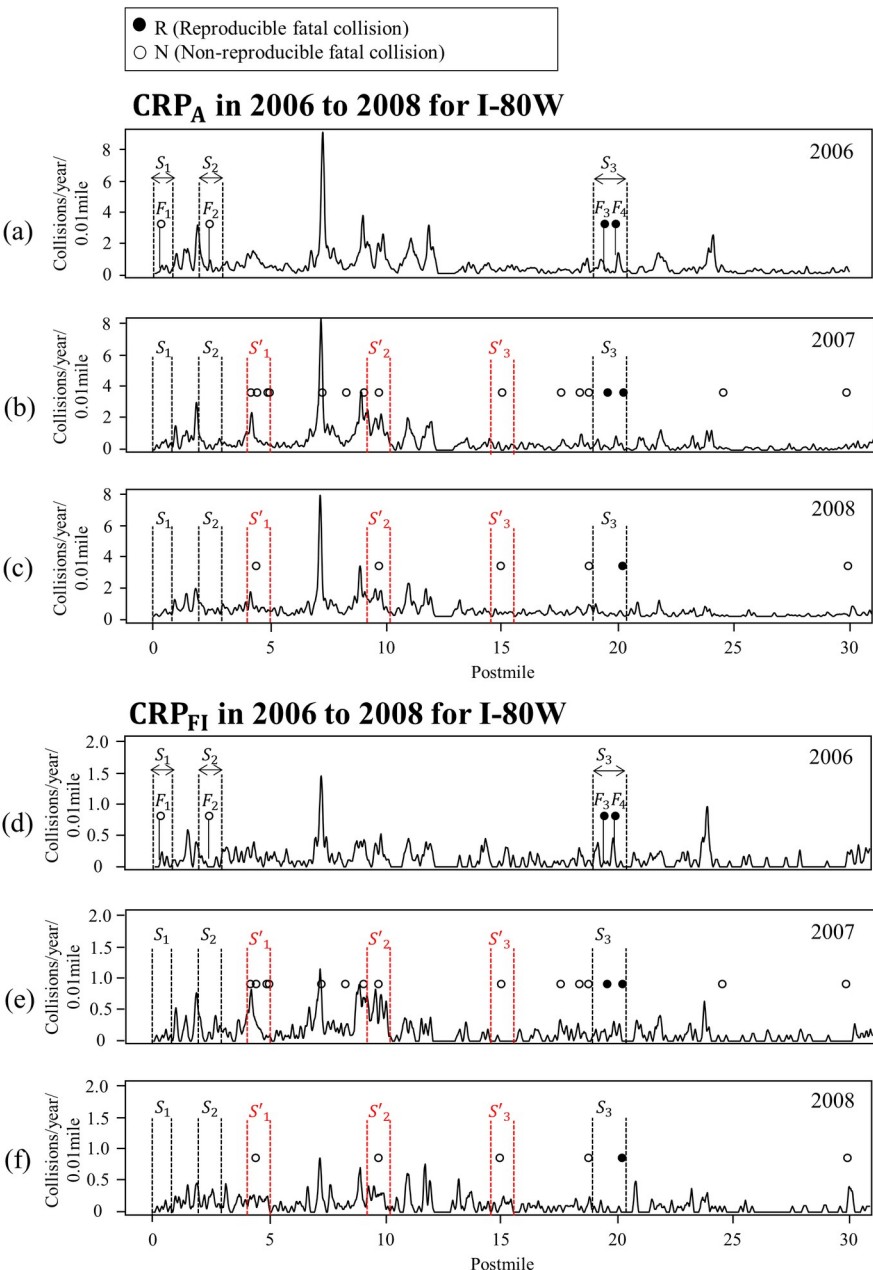

**Fig 2.** Fatal collision locations with $CRP_A$ in (a) 2006, (b) 2007, and (c) 2008, and fatal collision locations with $CRP_{FI}$ in (d) 2006, (e) 2007, and (f) 2008.

2006 and the validation years are 2007 and 2008. Fig 2 also shows a plot of the CRP, which estimates the true collision risk based on spatial patterns of collisions by filtering out statistical fluctuation, with the $R$ and $N$. The peaks in the CRP plot indicate the locations of local-collision concentrations. Fig 2A through 2F show plots of the CRPs constructed using traffic collision data, $CRP_A$, and those constructed using FI collision data, $CRP_{FI}$, observed along I-80W from 2006 to 2008. The black and white circles in Fig 2A and 2D indicate the locations of fatal collisions, respectively, with four of the ten fatal collisions observed during the reference year indexed from $F_1$ to $F_4$. $S_1$, $S_2$, and $S_3$ in the figures indicate site locations, with the site

boundaries (dotted vertical lines in Fig 2) determined by selecting $l/2$ upstream and down-stream of the location of the fatal collision in 2006. When another fatal collision had occurred within $l$, the boundary of the site was expanded by combining the $l$ value with respect to the additional fatal collision. In this study, 1.0 mile was used as the $l$ value, considering the potential for error in the reported collision location introduced when milepost information was entered into the traffic collision report [14] and due to the stochastic nature of the driver reaction to the causative factor of a fatal collision [25]. With $l = 1.0$ mile in our study sites, the lengths of the fatal collision sites ranged from 1 mile to 1.97 miles. We note how additional fatal collisions occurred in subsequent years within $S_3$, whereas no additional fatal collisions occurred within $S_1$ or $S_2$, as shown in Fig 2. Based on these empirical data, sites that experienced additional fatal collisions in subsequent years were classified as $R$ and the others were classified as $N$. $S'_1$ to $S'_3$ in Fig 2B and 2E indicate the locations of fatal collisions in 2007, which is not a reference year, and additional fatal collisions that occurred in 2008. If the reference year is 2007 and the validation years include 2008, sites $S'_1$ to $S'_3$ would be classified as $R$. In other words, the locations classified as $R$ and $N$ would differ with the reference and validation years. In the later section, we evaluate the proposed method in various combinations of reference and validation years.

## Naïve Bayesian approach for detecting reproducible fatal collision sites ($R$)

Naïve Bayes is a simple technique for modeling classifiers that assumes conditional independence of the observed features. In our approach, the observed features are the occurrence of fatal collisions along the freeway route in the reference year. Therefore, our approach assumes that the occurrence of each fatal collision is conditionally independent given $D$, which is a continuous random variable indicating the locations of $R$. Using Bayes' theorem, the conditional probability (i.e., posterior probability) of a site being classified as $R$ at $D$, $P(D|F_{1:m})$, given the $m$ fatal collision locations along the route in the reference year, can be calculated as shown in Eq (1):

$$P(D|F_{1:m}) = \frac{P(D)P(F_{1:m}|D)}{P(F_{1:m})} \tag{1}$$

where $P(D)$ is the prior probability of a site being classified as $R$ at $D$, and $P(F_{1:m}|D)$ is the likelihood of the occurrence of fatal collisions at locations $F_1$ to $F_m$ given $R$ at $D$. Since the purpose of the proposed method is to rank sites based their PMP values, and not to develop an unbiased estimate of a site being classified as $R$, the numerator, $P(D)P(F_{1:m}|D)$, is used to rank sites that are likely to be classified as $R$. The likelihood, $P(F_{1:m}|D)$, can be decomposed as shown in Eq (2) with the assumption of the conditional independence of fatal collision occurrences:

$$P(D|F_{1:m}) \propto P(D)P(F_{1:m}|D) = P(D)\prod_{i=1}^{m} P(F_i|D) \tag{2}$$

where $\propto$ indicates proportionality. The proposed method estimates the PMP of the site being classified as $R$ at $D$, $P(D)P(F_{1:m}|D)$, based on both the observed locations of fatal collisions, $F_{1:m}$, and the CRP of the reference year. The sites are then prioritized based on their PMP values regarding the recommendation for in-depth investigation. The CRP is used as a prior probability of a site being classified as $R$ in the procedure for estimating the PMP.

## Estimation of prior probability

The CRP estimates the true collision risk based on spatial patterns by filtering out statistical fluctuation. The CRP can be constructed using various filtering techniques such as the weighted moving average technique [20, 21] and ensemble empirical mode decomposition [22]. The level

of filtering is an important parameter for constructing the CRP, which can be empirically determined by investigating the changes in the number of critical points where the CRP slope is zero [21] or the spectral attributes of the traffic collision data [22]. In this study, the CRPs were constructed using the process described by Chung et al. [20], which filters out random noise as much as possible without affecting the location of the CRP peaks. More details and numerical examples of CRPs are presented in Chung et al. [20] and Chung et al. [21]. The shape of a CRP indicates the underlying true risk profile, and its value measures the collision frequency per unit distance of roadway. A continuous CRP profile can be used to consider the spatial correlation of traffic collisions and the area under the CRP indicates the expected collision frequency considering the regression-to-the-mean (RTM) phenomenon [26].

The unit of the CRPs shown in Fig 2 is the number of collisions per 0.01 mile per year along the route. A visual inspection of $CRP_A$ in Fig 2A–2C reveals the reproducible patterns over the years, which are consistent with the findings from previous studies [20, 21, 26]. $CRP_{FI}$ in Fig 2D–2F provides a stronger indicator of fatal collisions than those constructed using PDO, injury, and fatal collisions, $CRP_A$ (i.e., all traffic collisions), as it can lead to fatality depending on the characteristics of the victim: injury collisions may later be changed to fatal collisions when the injured person dies within a certain period of time after the collision [27]. However, the overall reproducibility of $CRP_{FI}$ compared to $CRP_A$ (see Fig 2A–2C) is smaller. In this study, both $CRP_A$ and $CRP_{FI}$ were evaluated with respect to the prior probability of a site being classified as R. These two prior probabilities are both used in estimating the PMP of a site being classified as R, based on the naïve Bayes approach.

## Estimation of likelihood

Fatal collisions can occur without any external causative factors. However, there may also exist factors that contribute to causing fatal collisions. If such factors exist, one would expect to see fatal collisions occurring near the location in which there are unknown fatal-collision causative factors. To consider the average effect of unknown causative factors (i.e., unobserved heterogeneity along spatial dimensions) of all fatal-collision locations on each route, we assumed the likelihood of the occurrence of observed fatal collisions given R at D, L(D), which is a function of the distance between fatal-collision locations ($F_i$) and the R location (D), as shown in Eqs (3a) and (3b). The influence of proximity among the fatal collisions on a site being classified as R was modeled using Eq (3b), where $d_i$ is the distance between $F_i$ and D, which indicates that the closer D is to $F_i$, the greater the effect of those unknown causative factors of $F_i$. The value of the likelihood function is one if the distance between $F_i$ and D is zero, and its minimum value is zero when $F_i$ and D are far away from each other. Other linear and non-linear functions sharing these properties have been applied, but they all exhibit comparable performance. Therefore, we selected the simplest function that is easy to implement in practice, as shown in Eq (3b). The hypothesis underlying the estimation of likelihood is that the collision rate near R may not be high enough to be detected by other HCCL identification procedures [15], or the reproducible fatal collision may not have a high enough concentration of collisions, as shown in Fig 2.

$$L(D) = \prod_{i=1}^{m} P(F_i|D) \tag{3a}$$

$$P(F_i|D) = \frac{1}{1 + d_i^{\alpha}} = \frac{1}{1 + |F_i - D|^{\alpha}} \tag{3b}$$

The likelihoods of the observed fatal collisions occurring at locations $F_1$ to $F_{10}$ are estimated based on D, which is the location of R. This can be further explained with reference to Fig 3A–3D. $F_1$ to $F_{10}$ in Fig 3 show the fatal-collision locations that occurred on I-80W in 2006, and $P(F|D = 10)$ is

the likelihood of a fatal collision occurring at $F$ when the assumed location of $R$ is 10. Therefore, the peak in Fig 3A representing $P(F \mid D = 10)$ at $F = 10$ is 1 by definition. Likewise, Fig 3B and 3C show the likelihood of the fatal collision occurring at $F$ for locations $R = 40$ and 60, respectively.

The likelihood of all fatal collisions along the entire route, $L(D)$, were estimated by combining the individual likelihood of the fatal collision, $\prod_{i=1}^{m} P(F_i|D)$, where $m$ is the total number of fatal collisions in 2006, i.e., the reference year (see Eq (3a)). $D$ is incremented by 0.01 mile and ranged over the entire route. Fig 3D shows $\prod_{i=1}^{10} P(F_i|D)$ according to $D$ over the entire I-80W route. The $\alpha$ (see Eq (3b)) determines how much distance between fatal collisions and $D$ influences $P(F_i \mid D)$. A higher value of $\alpha$ results in a stronger likelihood of $D$ being near $F$. A lower value of $\alpha$ indicates a diminished effect of the distance from a reproducible fatal-collision location in determining $P(F_i \mid D)$. Fig 4 shows how different values of $\alpha$ affect the shape of $P(F_i \mid D)$. When $\alpha = 0$, the shape of $P(F_i \mid D)$ remains horizontal. Higher values of $\alpha$ bring the shape of $P(F_i|D)$ to a wide peak near $D$ and the value of $P(F_i|D)$ drops sharply if the distance from $D$ become more than $L_c$, whereas lower values of $\alpha$ bring the shape of $P(F_i|D)$ to a narrow peak and the value of $P(F_i|D)$ drops gently even if the distance from $D$ is larger than $L_c$. The black circles in Fig 4 show the value of $P(F_i|D)$ at the postmile = 26 according to $\alpha$, which indicates that the high value of $\alpha$ makes the value of $P(F_i|D)$ more sensitive to the distance from $D$.

## Estimation of a proxy measure of the posterior probability

The naïve Bayesian approach is used to estimate the PMP based on likelihood, $L(D)$, and prior probability, $P(D)$. In other words, the combined influence of the proximity of fatal collisions at

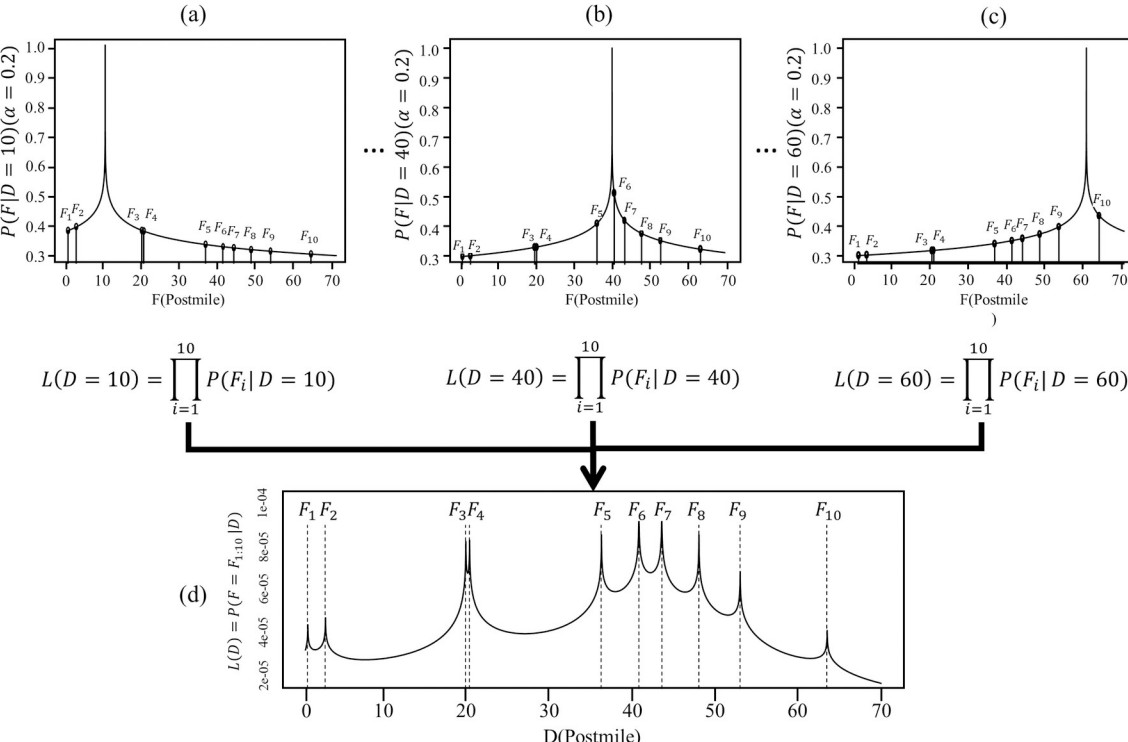

**Fig 3. Procedure for estimating the likelihood of fatal collisions occurring at locations $F_1$ to $F_{10}$ given $R$ at D on I-80W in 2006.** (a), (b), and (c) Individual likelihoods of fatal collisions at $D = 10$, $D = 40$, and $D = 60$, respectively. (d) Combined likelihood of all fatal collisions with D increasing by 0.01 mile increments along entire route.

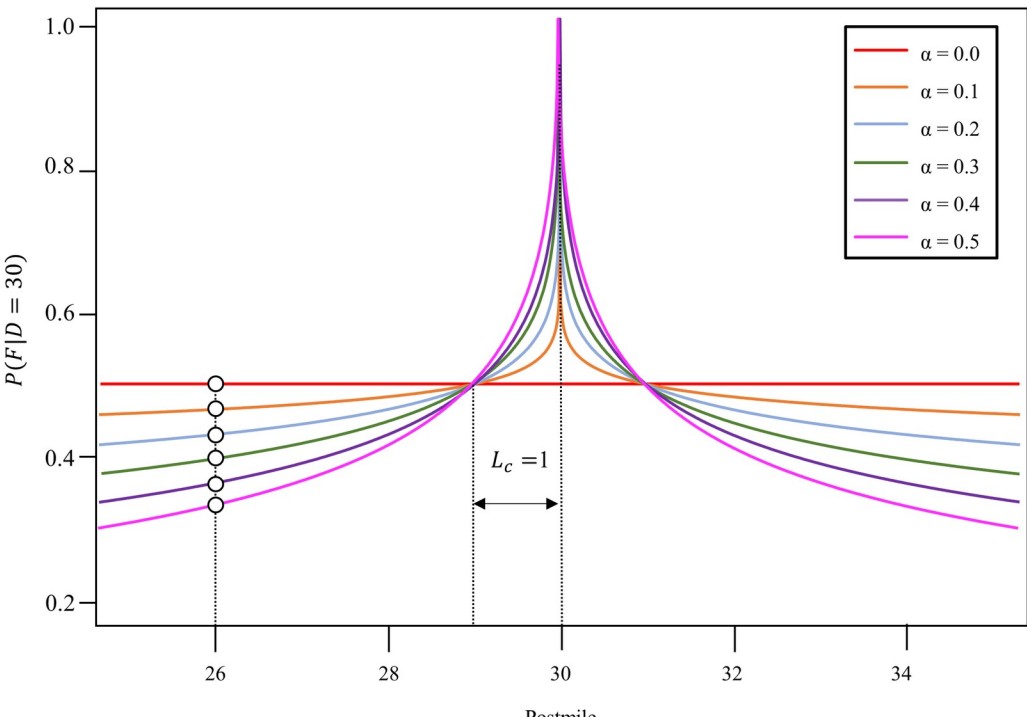

**Fig 4. Individual likelihoods of a fatal collision according to α when the R location is D = 30.**

a site is used as the likelihood, with the CRP as the indicator of the prior probability. $P(D)$ weighted by the $L(D)$ is used to estimate the PMP, as shown in Eq (4).

$$\text{PMP} = P(D)\prod_{i=1}^{m} P(F_i|D) = P(D)L(D) \tag{4}$$

Fig 5B and 5D show the resulting PMP estimated by $L(D)$ with α = 0.2 and the $P(D)$ based on $\text{CRP}_A$ (Fig 5A) and $\text{CRP}_{FI}$ (Fig 5C) together with the locations of the fatal collisions in the reference year. The PMPs of $S_1$, $S_2$, and $S_3$ being classified as $R$ are determined by the mean values of PMP within the site boundary, $\overline{\text{PMP}}$, which consist of the areas under the curve shaded in gray divided by the site length, as shown in Fig 5B and 5D. As shown in Fig 5B, the $\overline{\text{PMP}}$ of $S_3$ (0.0288) estimated using $\text{CRP}_A$ is much higher than those of $S_1$ (0.0093) and $S_2$ (0.0187). The $\overline{\text{PMP}}$ estimated using $\text{CRP}_{FI}$ exhibits the same pattern, with $S_3$ (0.0368) having the highest $\overline{\text{PMP}}$ value of being classified as $R$, as compared with $S_1$ (0.0112) and $S_2$ (0.0110). To determine the sites that should be classified as $R$, the estimated $\overline{\text{PMP}}$ of each fatal collision site was ranked based on the information available only in the reference year.

## Comparison of PMPs for all routes

To compare the PMPs of multiple routes, the PMP of a site being classified as $R$ for a single route (see Eq (4)) is rescaled to $\text{PMP}_R$, as shown in Eq (5):

$$\text{PMP}_R = [P(D) - SPF(D)]L(D)/\int_{D=s}^{D=e} L(D) \tag{5}$$

where $s$ and $e$ are the start- and end-points of the route, respectively. The $P(D)$ value (i.e.,

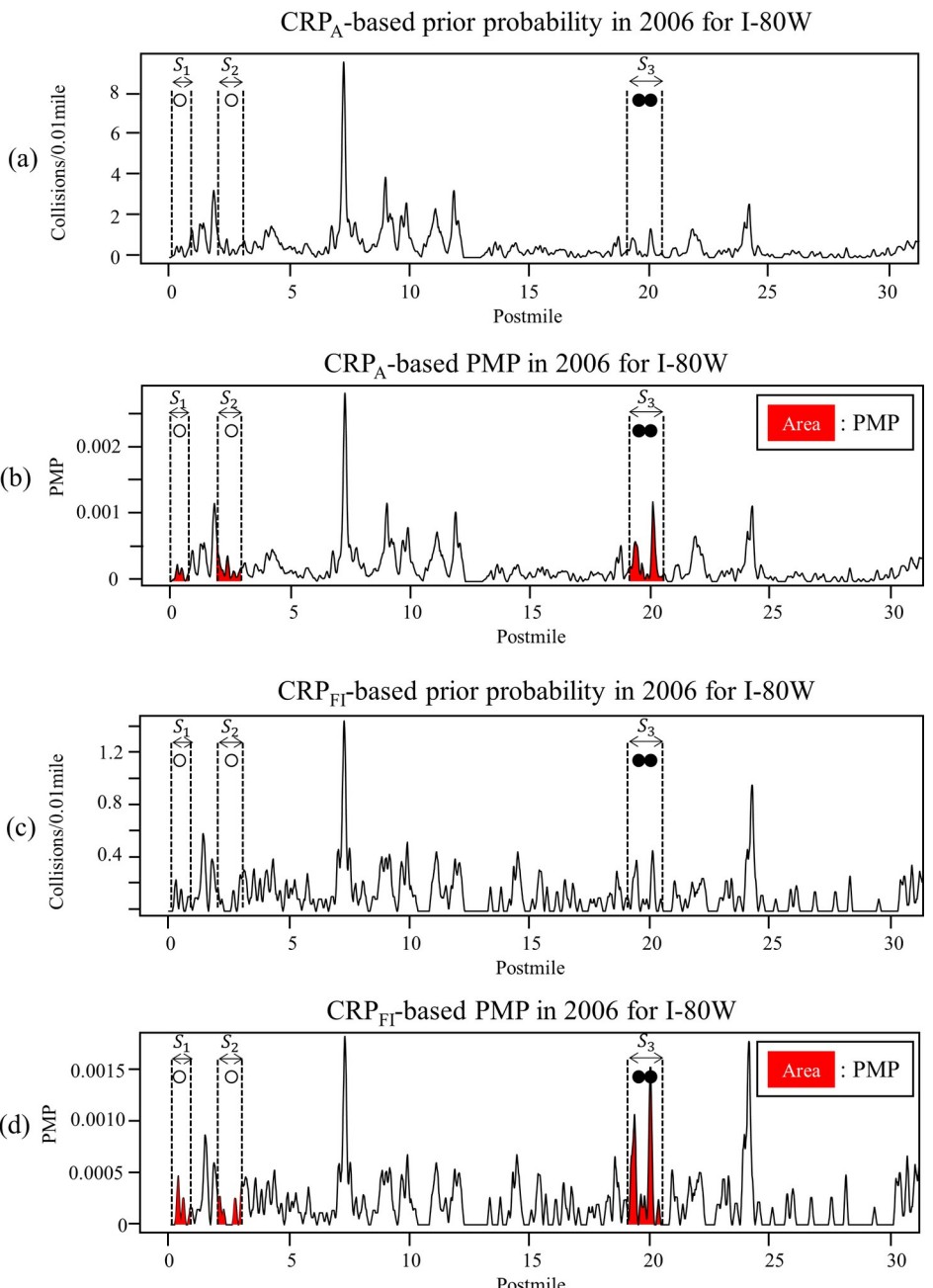

**Fig 5.** Results of naïve Bayesian approach to for I-80W in 2006: (a) $CRP_A$-based prior probability, (b) $CRP_A$-based PMP, (c) $CRP_{FI}$-based prior probability, and (d) $CRP_{FI}$-based PMP.

$CRP_A$ or $CRP_{FI}$) indicates the number of total or FI collisions per year per 0.01 mile. The $P(D)$ values of different routes cannot be directly compared because the collision causative factors, e.g., traffic volume, are different. The HSM [7] suggests use of the PSI, i.e., the difference between the expected and observed collision frequencies, as a safety measure to consider the collision causative factors in prioritizing sites to be investigated. This study applies an excess of CRP that is the difference between the CRP and SPF, which indicates the PSI [14]. The CRP can also consider the RTM phenomenon, which cause bias in estimating the benefits of safety

countermeasure for HCCLs, by filtering out random fluctuations in the collision frequency [26]. The value of $L(D)$ is a non-increasing function of the number of fatal collisions along the routes because $P(F_i|D)$ is always less than one (see Eq (3b)). To adjust for the biases introduced by the number of fatal collisions along each route, the $\mathrm{PMP_R}$ is estimated based on the rescaled likelihood, $L(D)/\int_{D=s}^{D=e} L(D)$.

To determine whether a site can be defined as $R$, the $\overline{\mathrm{PMP_R}}$ value is used, i.e., the mean value of $\mathrm{PMP_R}$ within the site boundary. Table 3 shows the $\overline{\mathrm{PMP_R}}$ value estimated by the proposed method for 22 fatal collisions associated with 19 sites on routes I-80W and I-80E in 2006, and the validation years are 2007 and 2008. Site lengths longer than 1.0 mile indicates the occurrence of an additional fatal collision within the boundary of the site, whereas site lengths shorter than 1.0 mile indicate that the fatal collision occurred near the start- or end-points of the route. The rows of Table 3 are arranged in descending order with respect to the $\overline{\mathrm{PMP_R}}$ estimated using $\mathrm{CRP_{FI}}$, with the value of $\boldsymbol{\alpha}$ being 0.3. Of the 19 fatal collision locations, five were classified as $R$, and four $R$s were included in the top five $\overline{\mathrm{PMP_R}}$ values. The overall performances of the six routes were evaluated together, and the findings are reported in the next section.

## Comparison of PMPs with SPF-based approaches

The SPF-based approach is a well-established procedures for detecting HCCLs [7, 8]. Conventional SPF-based approaches apply the empirical Bayes (EB) method, which estimates the expected number of collisions based on the dispersion of an SPF. As a safety measure, the PSI is used, i.e., the difference between expected crash frequency based on an EB adjustment and the SPFs [14]. In recent years, to differentiate between collision severity levels to better identify HCCLs, researchers have proposed the calibration of SPF for FI collisions [11, 12] or fatal

**Table 3. Site rankings evaluated based on the $\overline{\mathrm{PMP_R}}$ values for I-80W and I-80E.**

| Route | Site ID | Beginning postmile(mile) | End postmile(mile) | Site length(mile) | $\overline{\mathrm{PMP_R}}$ by $\mathrm{CRP_{FI}}$ | R |
|---|---|---|---|---|---|---|
| I-80E | $S_{13}$ | 21.295 | 22.395 | 1.100 | 0.00859 | Y |
| I-80E | $S_{14}$ | 23.035 | 24.505 | 1.470 | 0.00545 | Y |
| I-80E | $S_{12}$ | 20.095 | 21.095 | 1.000 | 0.00463 | N |
| I-80E | $S_{11}$ | 17.195 | 18.195 | 1.000 | 0.00170 | Y |
| I-80W | $S_3$ | 19.075 | 20.525 | 1.450 | 0.00114 | Y |
| I-80W | $S_4$ | 35.515 | 36.515 | 1.000 | 0.00104 | N |
| I-80W | $S_5$ | 40.055 | 41.055 | 1.000 | 0.00091 | N |
| I-80W | $S_7$ | 47.335 | 48.335 | 1.000 | 0.00071 | N |
| I-80E | $S_{15}$ | 35.075 | 36.075 | 1.000 | 0.00064 | Y |
| I-80W | $S_8$ | 52.335 | 53.335 | 1.000 | 0.00045 | N |
| I-80W | $S_1$ | 0.095 | 0.915 | 0.820 | 0.00035 | N |
| I-80W | $S_6$ | 42.825 | 43.825 | 1.000 | 0.00017 | N |
| I-80E | $S_{19}$ | 70.025 | 71.025 | 1.000 | 0.00016 | N |
| I-80E | $S_{17}$ | 55.255 | 56.255 | 1.000 | 0.00011 | N |
| I-80W | $S_2$ | 2.055 | 3.055 | 1.000 | 0.00008 | N |
| I-80E | $S_{10}$ | 1.765 | 2.765 | 1.000 | 0.00005 | N |
| I-80W | $S_9$ | 62.885 | 63.885 | 1.000 | 0.00004 | N |
| I-80E | $S_{18}$ | 63.985 | 64.985 | 1.000 | 0.00002 | N |
| I-80E | $S_{16}$ | 51.105 | 52.105 | 1.000 | 0.00001 | N |

collisions [13]. Advanced statistical models that consider the unobserved spatial heterogeneity have also been used to improve the SPF [28, 29]. However, the infrequent occurrence of fatal collisions causes a large variance in estimations of their frequency, with distributions of collision frequency skewing excessively toward zero, resulting in erroneous estimations [30]. This incorrect estimation is only exacerbated when estimating PSI because the observed frequency is also so low. Other researchers have proposed the property damage only equivalents (PDOEs) that weight collision frequencies based on collision severity with respect to the associated economic loss [15, 17]. They developed the SPF using PDOEs as a dependent variable, to consider the collision severity levels.

Table 4 shows the methodological considerations of the proposed method compared with previous approaches. Excess of CRP, i.e., the difference between the CRP and SPF, enables consideration of both causative factors and RTM bias, which are addressed by HSM's traditional SPF-based approach. The spatial heterogeneity due to the spatial dependence of traffic collisions can be considered using the CRP [20]. The likelihood of the occurrence of fatal collisions proposed in this study weights the CRP based on Bayes' theorem to consider fatal collisions by estimating the average effect of the unknown causative factors of fatal collisions and this likelihood reflects their unobserved spatial heterogeneity [5] for detecting reproducible fatal collision locations.

## Findings

### Performance measures

The performance of the proposed method in classifying sites as *R* based only on the data observed in the reference year was compared with that of a procedure that investigates fatal collision sites at random, i.e., the ad hoc site investigation procedure (e.g., sites may have been

**Table 4. Comparison of the methodological considerations of conventional SPF-based approaches and the proposed method.**

| Approach | Research | Methodological consideration | | | |
|---|---|---|---|---|---|
| | | Causative factors (method) | RTM (method) | USH (method) | Severity (target) |
| SPF calibrated to all collisions | [7] | ✓ | ✓ | - | - |
| | | (SPF) | (EB) | | |
| | [28] | ✓ | - | ✓ | - |
| | | (SPF) | | (GWNBR) | |
| | [29] | ✓ | - | ✓ | - |
| | | (SPF) | | (RPNBR) | |
| SPF calibrated to specific severities | [15,17] | ✓ | - | - | ✓ |
| | | (SPF) | | | (PDOEs) |
| | [11,12] | ✓ | - | - | ✓ |
| | | (SPF) | | | (FI) |
| | [13] | ✓ | - | - | ✓ |
| | | (SPF) | | | (Fatal) |
| SPF calibrated to all collisions with CRP method | [26] | ✓ | ✓ | ✓ | - |
| | | (SPF) | (CRP) | (CRP) | |
| $PMP_R$ | Proposed method | ✓ | ✓ | ✓ | ✓ |
| | | (CRP, Likelihood) | (Fatal) | (SPF) | (CRP) |

*Note*: Parentheses indicate the method or target for consideration in each column; PDOEs is property damage only equivalents; GWNBR is geographically weighted negative binomial regressions; RPNBR is random parameter negative binomial regressions; USH is unobserved spatial heterogeneity.

investigated based on the order in which the fatal collisions occurred within a fiscal year or on the number of parties involved). Fig 6 shows the performance of the proposed method in detecting reproducible fatal collision sites for the six routes with 2006 as the reference year and 2007 and 2008 as the validation years. The $x$-axis represents the number of sites investigated by the proposed method (i.e., top $x$ sites in the order of PMP) and random selection. The y-

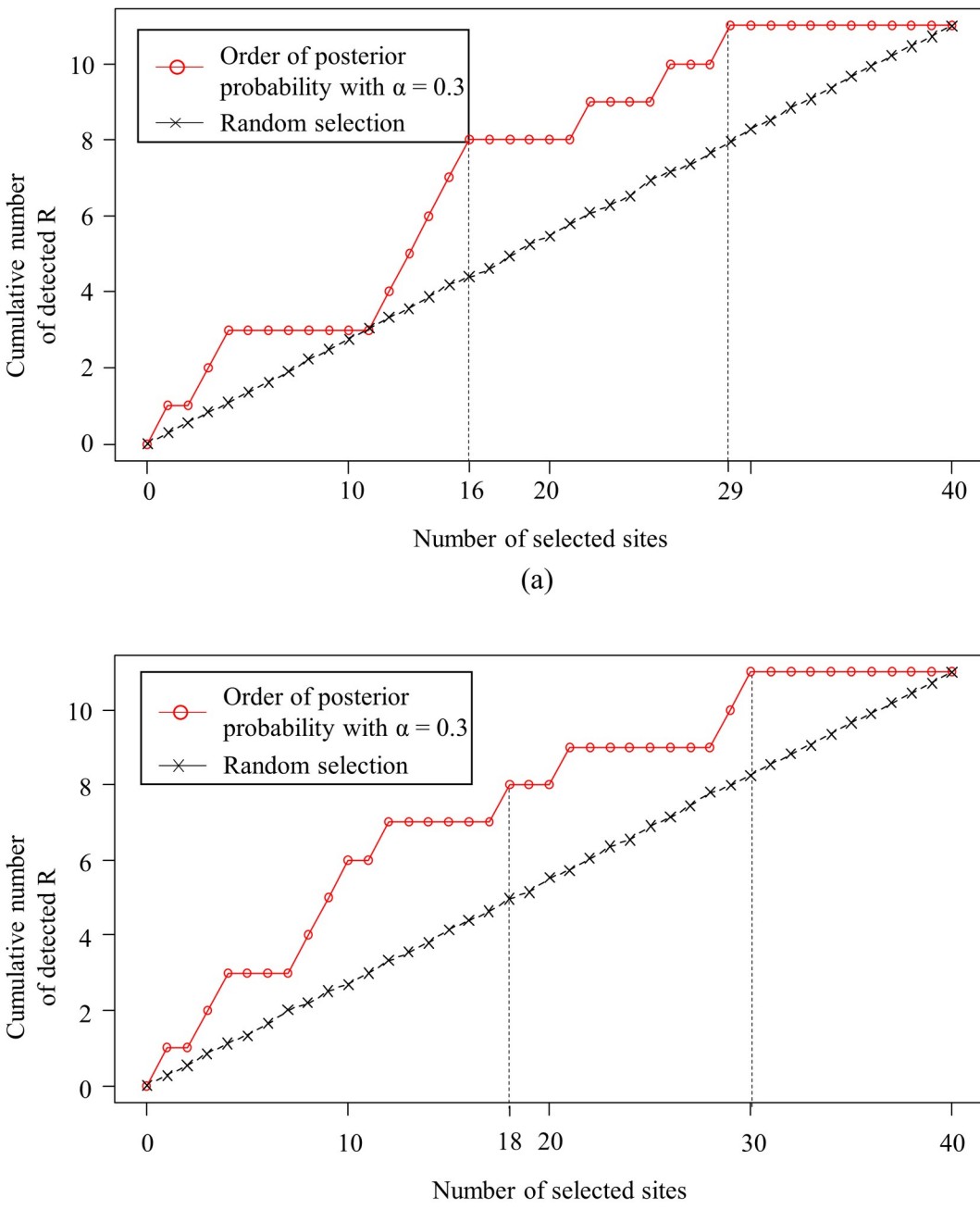

**Fig 6.** Performance in detecting reproducible fatal collision sites on six routes with $\alpha = 0.3$: (a) $CRP_A$ as a prior probability and (b) $CRP_{FI}$ as a prior probability.

axis represents the cumulative number of true $R$s identified with respect to the number of sites recommended for investigation. The dotted black line marked with x symbols in Fig 6 shows the performance in classifying sites as $R$ with no consideration of the likelihood and prior probability (i.e., random selection). The solid red line with circles shows the performance in classifying sites as $R$ based on $\overline{PMP}_R$. The $\overline{PMP}_R$ value was estimated based on the $CRP_A$ (see Fig 6A) and $CRP_{FI}$ (see Fig 6B), with the likelihood function $\alpha = 0.3$. The curves in Fig 6 provide information similar to the receiver operating characteristics (ROC), which are used to evaluate classification models [31]. If there are $n$ reproducible fatal collision sites among the $N$ fatal collision sites at a study site, the number of sites investigated is $x_i$, and the corresponding cumulative number of sites classified as $R$ is $y_i$, the false negative (i.e., not detecting a true $R$) is $n - y_i$, false positive (i.e., flagging a site as $R$ when is not) is $x_i - y_i$, true positive (TP) is $y_i$, and true negative is $N - n - x_i + y_i$. The recall and precision are calculated as shown in Eqs (6) and (7). Note that precision and recall vary with $x_i$ as they have a trade-off relationship. Fig 6 shows the trade-off between precision and recall with respect to $x_i$ and the area under this curve (AUC) indicates the overall performance of the proposed model. The fact that the AUC of the proposed method (i.e., the dotted red lines) is larger than those obtained by random selection (i.e., the dotted black lines) shows that the proposed method outperforms random selection.

$$\text{Recall} = \frac{TP}{TP + FN} = \frac{y_i}{n} \tag{6}$$

$$\text{Precision} = \frac{TP}{TP + FP} = \frac{y_i}{x_i} \tag{7}$$

Based on the above evaluation, the practical impact of the proposed method is as follows. In Fig 6, 11 $R$s were classified out of the total 40 fatal collision sites along the six routes. Note that if the government agency uses the proposed method with $CRP_A$ (see Fig 6A), it will have to investigate 16 of 40 sites (40.0%) to identify 8 of the 11 reproducible fatal collision sites (72.7%). To identify all $R$s, 29 sites will have to be investigated by $CRP_A$. Using $CRP_{FI}$ (see Fig 6B), the agency will also have to select 18 sites to identify 8 out of 11 $R$s, but 30 sites will need to be investigated to identify all $R$s.

## Comparison of performance with SPF-based approach

To compare the proposed method with the SPF-based approach that uses the EB method [32], we applied recently developed SPFs to the study site [14] to detect reproducible fatal collision sites. SPF-based approaches prioritize fatal collision locations using PSI, which is the difference between the expected and observed collision frequencies. Kwon et al. [14] developed SPFs for each type of roadway group, considering rural or urban, arterial or highway, and the number of lanes, and their SPFs were found to better fit the data for the study sites than the SPFs currently used by Caltrans. As an explanatory variable, Kwon's SPF uses the average annual daily traffic (AADT) volume, although other additional variables such as shoulder width and horizontal curves [11] can improve the SPF performance. This study uses Kwon's SPF for two reasons: i) Caltrans and practitioner currently uses this simple type of SPF based on AADT and roadway group [14] because developing its own database to keep track of the values of additional roadway information is cost-prohibitive and ii) only traffic collision and AADT data are required for practical applications like the proposed method. We used the SPF calibrated for FI crashes in each highway group and estimated the PSI for fatal collision sites using the EB method. Comparison results of the proposed method and SPF-based approach are provided in the following subsections with sensitivity analyses.

## Sensitivity analysis of likelihood parameter

The sensitivity of the likelihood parameter $\alpha$, which adjusts the influence of the proximity of fatal collisions, is examined by applying $\alpha$ values from 0 to 0.7 for the six routes in the same reference and validation years as those in Fig 6. The sensitivity was evaluated by the area under the receiver operating characteristics (AUROC) curve, which is a well-known performance measure for classification models [31]. The AUROC shows the competitiveness of the classification model as compared with random guessing, similar to the area under the red and black lines in Fig 6.

An AUROC value higher than 0.5 means the model outperforms random guessing, with the best score possible being 1.0. Fig 7 shows the performance evaluation with $\alpha$ for the $CRP_A$ and $CRP_{FI}$ priors, both of which outperformed random guessing with AUROC values higher than 0.5. The proposed method also outperforms the SPF-based approach, which obtained a 0.53 AUROC value only slightly better than random guessing. For a lower value of $\alpha$, which indicates a weak influence of fatal collision locations on the classification of sites as $R$, the $CRP_{FI}$ outperforms the $CRP_A$, although the $CRP_A$ performs better for a higher value of $\alpha$. The best performances were obtained by the $CRP_{FI}$ prior, with a 0.75 AUROC value for $\alpha = 0.3$ and 0.4 and the $CRP_A$ prior with a 0.75 AUROC value for $\alpha = 0.6$. The performances of $CRP_A$ and $CRP_{FI}$ with various $\alpha$ values were comparable, which shows that the proposed method is robust with respect to the parameter $\alpha$ and the prior probability.

## Sensitivity analysis for reference and validation years

Reproducible fatal collision sites are classified based on the occurrence of additional fatal collisions in the validation years near where the fatal collision occurred in the reference year. Since traffic collisions are rare and random events, there could be multiple reference and validation years. Although a multiyear time period takes advantage of the RTM phenomenon [32], it causes within-period variation in the presence of unobserved heterogeneity [30]. To evaluate the generalization performance of the proposed method with respect to the reference and validation years, we evaluated our method under various conditions, as shown in Table 5. For example, sites are classified as $R$ in Condition 1 when additional fatal collisions occurred in

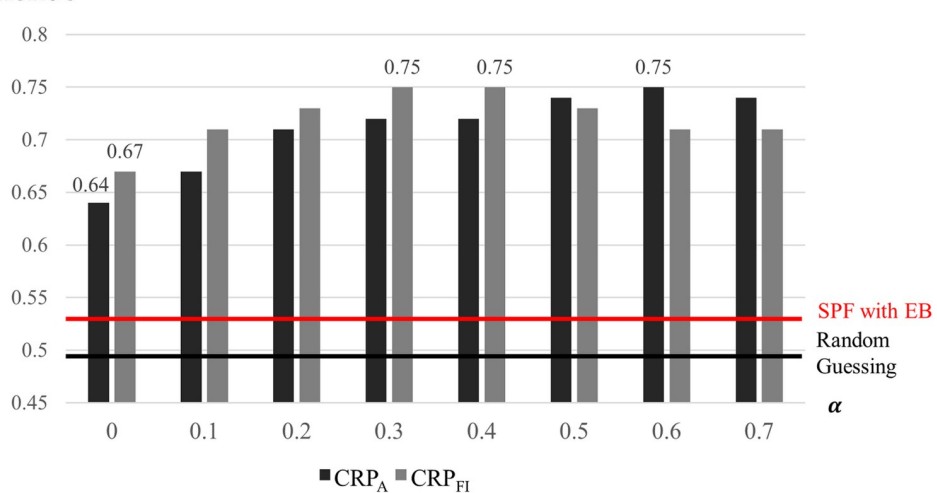

**Fig 7. Performances in detecting reproducible fatal collision sites for the six routes using the proposed method according to α and CRP priors, compared to using the SPF-based approach.**

**Table 5. Performance in detecting reproducible fatal collision sites on six routes with α = 0 to 0.6 with CRPFI priors for various reference and validation years.**

| Condition | | 1 | 2 | 3 | 4 | 5 |
|---|---|---|---|---|---|---|
| Reference years | | 2006 | 2005 | 2005, 2006 | 2004, 2005 | 2004, 2005, 2006 |
| Validation years | | 2007, 2008 | 2006, 2007 | 2007, 2008 | 2006, 2007 | 2007, 2008 |
| Prior | | $CRP_{FI}$ | $CRP_{FI}$ | $CRP_{FI}$ | $CRP_{FI}$ | $CRP_{FI}$ |
| | | AUROC | | | | |
| FI-SPF approach with EB method | | 0.53 | 0.56 | 0.57 | 0.47 | 0.59 |
| α | 0 | 0.67 | 0.63 | 0.68 | 0.65 | 0.62 |
| | 0.1 | 0.71 | 0.64 | 0.70 | **0.66** | 0.65 |
| | 0.2 | 0.73 | **0.65** | **0.71** | 0.64 | **0.66** |
| | 0.3 | **0.75** | 0.64 | 0.70 | 0.63 | 0.64 |
| | 0.4 | **0.75** | 0.63 | 0.68 | 0.60 | 0.62 |
| | 0.5 | 0.73 | 0.62 | 0.67 | 0.57 | 0.61 |
| | 0.6 | 0.71 | 0.61 | 0.65 | 0.56 | 0.61 |

2007 or 2008 near where the fatal collision occurred in 2006. In the case of Condition 3, among the fatal collision locations in 2005 or 2006, sites are classified as *R* when additional fatal collisions occurred at these locations in 2007 or 2008. If there are multiple reference years, the CRP, i.e., the prior probability of sites being classified as *R*, is constructed from the sum of the collisions in multiple years.

Table 5 shows the performance of the proposed method for various conditions in classifying sites as *R*. The proposed method with the $CRP_{FI}$ prior was evaluated with respect to the likelihood parameter α, and these performances were compared with those of the SPF-based approach using the EB method. For all conditions, the proposed method outperforms the SPF-based approach with the EB method. In condition 4, the performance of the SPF-based approach was worse than random selection (i.e., 0.5 AUROC), which means that the SPF calibrated using FI collisions cannot contribute to the classification of a site as *R*. In each condition, the best AUROC values ranged from 0.65 to 0.75, with the likelihood parameter α values ranging from 0.1 to 0.3. These results reveal that although the performance of the proposed method could vary with the data used to classify sites as *R*, it outperforms random selection and the SPF-based approach in detecting *R*s.

Although the proposed method successfully detected the reproducible fatal collision locations based on the $PMP_R$ estimated by $CRP_A$ and $CRP_{FI}$, the causative factors for those locations requires further exploration to apply effective countermeasures for roadway safety. We investigated the primary collision factors for 145 fatal collisions that occurred on the six routes from 2006 to 2008 and divided the fatal-collision sites into reproducible and non-reproducible locations. The reference year was 2006 and the validation years were 2007 and 2008. Fig 8 shows distributions of the primary collision factors for the reproducible and non-reproducible fatal collision locations, for which a chi-square test was conducted to determine if there is a difference in the frequency of the causative factors of these traffic collisions. Because the chi-square test assumes that no more than 20% of the cell counts are less than five [33], we aggregated three low-incidence collision factors—"Other Violations," "Improper Driving," and "Other Than Driver,"—as "Other Violations." The P-value of the chi-square test was 0.87, which indicates no significant difference in the distributions of primary collision factors between the reproducible and non-reproducible fatal collision locations. This may mean that the primary collision factor cannot be used to classify reproducible fatal collision locations.

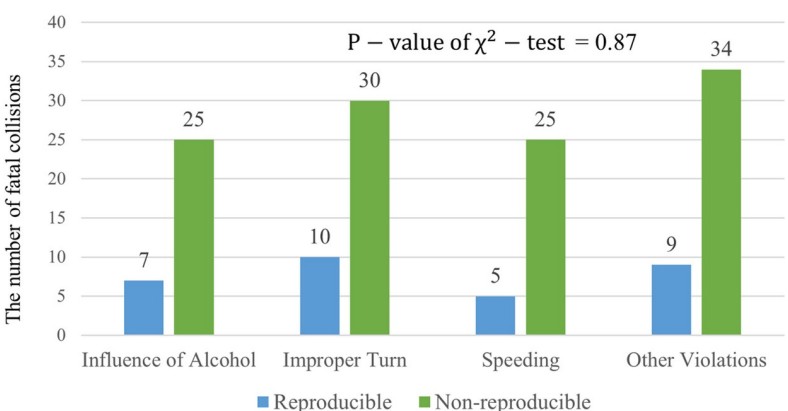

**Fig 8. Distributions of primary collision factors for reproducible and non-reproducible fatal collision sites on the six routes from 2006 to 2008.**

## Concluding remarks

State agencies are mandated to investigate the locations of fatalities and serious injuries. However, most of existing methods for detecting high-collision-concentration locations do not consider the severity of the collisions, which results in agencies focusing on sites with a high frequency of property damage only collisions while ignoring sites with higher percentages of injury and fatal collisions. Several previous studies have proposed methods for considering the severity of collisions by weighting them with respect to their associated economic losses or the use of a safety performance function (SPF) calibrated to fatal or fatal and injury collisions. However, these approaches can yield a high false-positive rate because fatal collisions that occur randomly and rarely due to the fault of the driver can still dwarf the weight of fatal collisions or cause high variance in the SPF.

To better utilize limited government resources while improving the safety of sites where fatal collisions occur, we developed a method for estimating the proxy measure of the posterior probability (PMP) of a site being classified as *R* (i.e., a site that is likely to experience an additional fatal collision in the vicinity in the subsequent year). The peaks of the continuous risk profile (CRP), which are reproducible over the years, were used as indicators of the prior probability of reproducible collision locations, and the likelihood of the occurrence of fatal collision was computed by the spatial distribution function for fatal collisions. Then, the likelihood and the prior values were multiplied and normalized to obtain the $PMP_R$ values, which can be compared for different routes. The empirical evaluations indicated that the $PMP_R$ of a site being classified as *R* can successfully identify sites as *R* compared with a random selection of sites or the SPF-based approach with the EB method. This finding can assist a state agency to better allocate its limited resources. A sensitivity analysis of the proposed method was also conducted to determine its robustness to some variations of the method, such as the choice of the prior indicator between $CRP_A$ and $CRP_{FI}$, the influence parameter, $\alpha$, and the data used for classifying reproducible fatal collision locations. The results indicate that the proposed method achieves robust performance regardless of these parameters and conditions.

Although advanced vehicle safety technology, advanced highway design, and surrogate safety-measures have developed from the past, the number of total and fatal collisions is still rising, which indicates that the HCCL still remains in the freeway. The proposed method does not find the HCCL based on the complex human and environmental factors but finds the recurrent fatal collision locations using a probabilistic approach based on the locations and

number of collisions. In other words, the proposed method can be applied to the recent data since it does not depend on the complex influential factors that change over time. Nevertheless a comparison of the proposed method with other methods using a recent dataset would be further research subject to verify the generalized performance. As the form of the likelihood function was assumed in the present study, evaluating the performance of different likelihood functions will be the subject of future study. This study estimated the likelihood of fatal collision based only on the spatial distribution of fatal collision locations, which is a great advantage for practical application in terms of data availability. However, if significant fatal-collision causative factors are identified, those factors could also be considered in the likelihood function. Therefore, evaluating the impact of other causative factors such as weather, roadway geometry, and traffic characteristics in classifying sites as $R$ [34–37] and other likelihood functions that consider those factors will be the subjects of future study. The authors also plan to further evaluate the performance of the proposed method along rural highways or urban area where the characteristics of fatal collisions can differ from those of this study [38].

## Supporting information

**S1 Dataset. The raw data employed in this study.**
(ZIP)

## Author Contributions

**Conceptualization:** Oh Hoon Kwon.

**Data curation:** Eui-Jin Kim.

**Formal analysis:** Eui-Jin Kim.

**Funding acquisition:** Dong-Kyu Kim.

**Investigation:** Eui-Jin Kim, Oh Hoon Kwon, Koohong Chung.

**Methodology:** Oh Hoon Kwon.

**Project administration:** Shin Hyoung Park.

**Supervision:** Koohong Chung.

**Validation:** Eui-Jin Kim, Koohong Chung.

**Writing – original draft:** Oh Hoon Kwon.

**Writing – review & editing:** Eui-Jin Kim, Shin Hyoung Park, Dong-Kyu Kim, Koohong Chung.

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
