## [Decision Letter · Decision Letter 0]

18 Mar 2021

PONE-D-21-03606

Application of Naïve Bayesian Approach in Detecting Reproducible Fatal Collision Locations on Freeway

PLOS ONE

Dear Dr. Kwon,

Thank you for submitting your manuscript to PLOS ONE. After careful consideration, we feel that it has merit but does not fully meet PLOS ONE’s publication criteria as it currently stands. Therefore, we invite you to submit a revised version of the manuscript that addresses the points raised during the review process.

Please try to revise your manuscript and respond to all the reviewers' comments.

We look forward to receiving your revised manuscript.

Kind regards,

Quan Yuan, Ph.D.

Academic Editor

PLOS ONE

Journal Requirements:

2. Please add information in both the Methods section and the Data availability statement on how the real dataset was obtained, and whether the authors have permission to share this openly with the publication.

3. We note that Figure in your submission contains map images which may be copyrighted. All PLOS content is published under the Creative Commons Attribution License (CC BY 4.0), which means that the manuscript, images, and Supporting Information files will be freely available online, and any third party is permitted to access, download, copy, distribute, and use these materials in any way, even commercially, with proper attribution. For these reasons, we cannot publish previously copyrighted maps or satellite images created using proprietary data, such as Google software (Google Maps, Street View, and Earth). For more information, see our copyright guidelines: http://journals.plos.org/plosone/s/licenses-and-copyright.

(1) You may seek permission from the original copyright holder of Figure 1 to publish the content specifically under the CC BY 4.0 license. 

Reviewers' comments:

Reviewer's Responses to Questions

**Comments to the Author**

1. Is the manuscript technically sound, and do the data support the conclusions?

Reviewer #1: Yes

Reviewer #2: Yes

2. Has the statistical analysis been performed appropriately and rigorously? 

Reviewer #1: Yes

Reviewer #2: Yes

3. Have the authors made all data underlying the findings in their manuscript fully available?

Reviewer #1: Yes

Reviewer #2: Yes

4. Is the manuscript presented in an intelligible fashion and written in standard English?

Reviewer #1: Yes

Reviewer #2: Yes

5. Review Comments to the Author

Reviewer #1: This study developed systematic ways of detecting reproducible fatal collision locations (R) using the naïve Bayes approach and a continuous risk profile (CRP) that estimates the true collision risk by filtering out random noise in the data. The results can provide policy support for reducing the risk of traffic accidents.

Review result：Accept

There are several questions that the author should explain or add: 1. The research data in this paper are traffic accident data of five similar highways in the United States from 2004 to 2008, and the traffic data are too old to have low reference significance for traffic accident prediction and safety improvement under modern traffic conditions. Also in the article "In 2016 alone, there were 1,389 fatal collisions on 15,181 miles of the state highway systems in California [5]. " This does not affect the content of the article, but it is questionable to use data from the past 10 years to argue and analyze the existing traffic phenomenon.

2. The quality of the pictures in this article is vague and the content is difficult to identify.

3. The vehicle technology conditions, road conditions, and traffic conditions of existing traffic activities have developed radically changed, how do the research results of the paper reflect the significance of guidance for reality?

Reviewer #2: In order to address the issue a high false-positive rate to identify the high-collision-concentration locations with fatal collisions in previous studies, the paper developed a method for estimating the proxy measure of the posterior probability and compared the improvements with the SPF-based approach. Their work was detailed and complete.

However, the quality of the pictures in the manuscripts needs to be improved. Resolution of most images are too low.

6. PLOS authors have the option to publish the peer review history of their article (what does this mean?). If published, this will include your full peer review and any attached files.

Reviewer #1: No

Reviewer #2: No

---

## [Author Response · Author response to Decision Letter 0]

12 Apr 2021

We would like to express our sincere gratitude for the comments and suggestions from the reviewers. In the "Response to Reviewers" and "Revised manuscripts with track changes" files, essential modifications and response to each comment were made based on comments of editor and reviewers.

---

## [Decision Letter · Decision Letter 1]

5 May 2021

Application of Naïve Bayesian Approach in Detecting Reproducible Fatal Collision Locations on Freeway

PONE-D-21-03606R1

Dear Dr. Kwon,

We’re pleased to inform you that your manuscript has been judged scientifically suitable for publication and will be formally accepted for publication once it meets all outstanding technical requirements.

Kind regards,

Quan Yuan, Ph.D.

Academic Editor

PLOS ONE

Additional Editor Comments (optional):

Reviewers' comments:

Reviewer's Responses to Questions

**Comments to the Author**

1. If the authors have adequately addressed your comments raised in a previous round of review and you feel that this manuscript is now acceptable for publication, you may indicate that here to bypass the “Comments to the Author” section, enter your conflict of interest statement in the “Confidential to Editor” section, and submit your "Accept" recommendation.

Reviewer #1: All comments have been addressed

Reviewer #2: All comments have been addressed

2. Is the manuscript technically sound, and do the data support the conclusions?

Reviewer #1: Yes

Reviewer #2: Yes

3. Has the statistical analysis been performed appropriately and rigorously? 

Reviewer #1: Yes

Reviewer #2: Yes

4. Have the authors made all data underlying the findings in their manuscript fully available?

Reviewer #1: Yes

Reviewer #2: Yes

5. Is the manuscript presented in an intelligible fashion and written in standard English?

Reviewer #1: Yes

Reviewer #2: Yes

6. Review Comments to the Author

Reviewer #1: (No Response)

Reviewer #2: The authors had addressed the problem I mentioned. Some minor improvements would be make the paper better. the abbreviations in the manuscript are too simple or not well explained, especially in the abstract. R representing reproducible fatal collision locations are too simple and SPF has many different complete expressions.

7. PLOS authors have the option to publish the peer review history of their article (what does this mean?). If published, this will include your full peer review and any attached files.

Reviewer #1: No

Reviewer #2: No

---

## [Editor Report · Acceptance letter]

7 May 2021

PONE-D-21-03606R1 

Application of Naïve Bayesian Approach in Detecting Reproducible Fatal Collision Locations on Freeway 

Dear Dr. Kwon:

I'm pleased to inform you that your manuscript has been deemed suitable for publication in PLOS ONE. Congratulations! Your manuscript is now with our production department. 

Kind regards, 

on behalf of

Dr. Quan Yuan 

Academic Editor

PLOS ONE